# A Competitive “On-Off-Enhanced On” AIE Fluorescence Switch for Detecting Biothiols Based on Hg^2+^ Ions and Gold Nanoclusters

**DOI:** 10.3390/bios13010035

**Published:** 2022-12-27

**Authors:** Shuqi Li, Yuqi Wan, Yu Li, Jinghan Liu, Fuwei Pi, Ling Liu

**Affiliations:** 1State Key Laboratory of Food Science and Technology, School of Food Science and Technology, Jiangnan University, Wuxi 214122, China; 2Collaborative Innovation Center of Food Safety and Quality Control in Jiangsu Province, Jiangnan University, Wuxi 214122, China; 3Wuxi Institute of Technology, Wuxi 214122, China

**Keywords:** aggregation-induced emission, fluorescence enhancement, detection, biothiols, sensitive

## Abstract

In this study, a novel “on-off-enhanced on” approach to highly sensitive rapid sensing of biothiols was developed, based on competitive modulation of gold nanoclusters (AuNCs) and Hg^2+^ ions. In our approach, the AuNCs were encapsulated into a zeolite imidazole framework (ZIF) for predesigned competitive aggregation-induced luminescence (AIE) emission. To readily operate this approach, the Hg^2+^ ions were selected as mediators to quench the fluorescence of AuNCs. Then, due to the stronger affinities between the interactions of Hg^2+^ ions with -SH groups in comparison to the AuNCs with -SH groups, the quenched probe of AuNCs@ZIF-8/Hg^2+^ displayed enhanced fluorescence after the Hg^2+^ ions were competitively interacted with -SH groups. Based on enhanced fluorescence, the probe for AuNCs@ZIF-8/Hg^2+^ had a sensitive and specific response to trace amounts of biothiols. The developed fluorescence strategy had limit of quantification (LOQ) values of 1.0 μM and 1.5 μM for Cys and GSH molecules in serum, respectively. This competitive AIE strategy provided a new direction for developing biological probes and a promising method for quantifying trace amounts of biothiols in serum. It could promote progress in disease diagnosis.

## 1. Introduction

Biothiols, as essential signaling molecules, e.g., cysteine (Cys) and glutathione (GSH), play important roles in physiological activities [1]. Several studies have confirmed that abnormal levels of biothiols were highly associated with neurotoxicity and cardiovascular disease [2,3,4]. For example, a high level of homocysteine is a reliable risk factor for hypertension, stroke and heart attack [5,6]. Significantly reduced GSH levels in serum can be used as diagnostic signs of early Parkinson’s disease. Levels of GSH are related to neurodegenerative diseases, diabetes, HIV infection and cancer [7,8,9]. In addition to the above diseases, sudden changes in Cys molecule levels are potentially associated with liver damage, skin diseases and Alzheimer’s disease [10,11,12]. Therefore, the necessary point-of-care-testing (POCT) techniques relating to evaluation of biothiol levels—especially visual approaches—have attracted more attention in analytical sciences.

Currently, traditional methods, such as capillary electrophoresis [13], high performance liquid chromatography [14], colorimetry [15] and mass spectrometry [16] are widely used in the evaluation of biothiols. However, expensive equipment, time-consuming operation and complex preparations highly limit onsite applications. Over the past decade, rapid fluorescence detection technology, based on various fluorescent probes, has developed rapidly, providing simple, robust, and fast sensing strategied for the quantitative analysis and monitoring of biological substances, biologically relevant ions and other trace substances [17,18,19]. Nanoprobes, as another alternative approach, have also been seen as promising strategies for rapid sensing of biothiols [20,21]. Among these, gold nanoclusters (AuNCs) have been marked as significant probes on fluorescence output due to their simple synthetic methods, good photostability, rapid sensing, chemical stability and biocompatibility [22]. Notwithstanding, to date, most synthetic techniques for AuNCs have displayed lower fluorescence quantum yields compared to common fluorescent dyes such as rhodamine [23,24]. Thus, a promising approach for modulating the fluor-features of AuNCs has become necessary and desirable for Au-based POCT techniques.

As a class of porous crystalline materials, zeolite imidazole frameworks (ZIF-MOFs) have been used to adsorb and separate various ions or applied as a carrier to co-encapsulate molecular probes, such as enzymes, carbon dots and gold nanomaterials [25,26,27,28]. Due to the confining holes of ZIF-MOFs, the encapsulated molecules usually generate more effective performances than those of which they are capable. For example, Jingtian Chi et al. [29] incorporated AuNCs with glucose oxidase (GO) into ZIF-8, which greatly improved the catalytic activity of the enzymes and the storage stability of the nanomaterials compared to the free GOx/AuNCs solution under the protection of ZIF-8 scaffold. For highly effective detection of tetracycline residues in milk, Alireza Khataee et al. [30] encapsulated both AuNCs and copper nanoclusters (CuNCs) into ZIF-8. Based on their construction, the quantum yields of BSA-AuNCs were improved fifteenfold due to restricted molecular motion and a 4.8 nM of the limit of detection was obtained. The significantly increased emission efficiency of AuNCs and the protective effects of the ZIF-MOFs shell couldpave a promising way forward for further application of AuNCs in hazards, specifically in molecule sensing and sensitive POCT evaluations.

In the present study, a novel “on-off-enhanced on” approach was developed, based on the competitive modulation of AuNCs in ZIF-8-MOF for highly sensitive rapid sensing of biothiols. As illustrated in Figure 1, in our strategy, the AuNCs were encapsulated into ZIF-8-MOF. Due to the self-aggregation of AuNCs, an intensive aggregation-induced luminescence was achieved. Differing from conventional AIE ideas, the aggregated luminescence was modulated and quenched through a pre-burying competitive factor of Hg^2+^ ions to develop a visual-sensing nanoprobe. In the presence of biothiols, the mediators of Hg^2+^ ions were competed from AuNCs, based on the formation of stronger Hg^2+^-S bonds, which resulted in the recovery and generation of enhanced fluorescence. A nanoprobe such as that devised could successfully respond to trace levels of biothiols in serum samples. Notably, the solid stability of free AuNCs also demonstrated great potential for sensing biothiols and could promote progress in disease diagnosis.

## 2. Experimental Section

### 2.1. Materials and Apparatus

Chloroauric acid trihydrate (HAuCl_4_·3H_2_O, purity > 99.9%) was supplied by Adamas Reagent Co., Ltd. (Shanghai, China). Glutathione reduced (GSH) was purchased from Innochem Co., Ltd. (Beijing, China). 2-Methylimidazole (2-MIM) was purchased from J&K Scientific Co., Ltd. (Shanghai, China). Zinc nitrate hexahydrate (Zn(NO_3_)_2_, purity > 99%), L-Cysteine(Cys), L-Aspartic acid (Asp), L-Arginine (Arg), L-Lysine (Lys), Glycine (Gly), L-Valine (Val), L-Isoleucine (Ile), L-Leucine (Leu), L-Histidine (His), L-Phenylalanine (Phe), L-Threonine (Thr), L-Proline (Pro), L-Alanine (Ala), L-Glutamic acid (Glu), L-Tryptophan (Trp), L-Serine(Ser), L-Methionine (Met), L-Glutamine (Gln) L-Asparagine (Asn) and tyrosine (Tyr) were purchased from Sinopharm Chemical Reagent Co., Ltd. (Shanghai, China).

All chemical reagents were AR-grade and used without further purification. The ultrapure water used throughout the experiments was obtained from a Millipore water purification system (18.2 MΩ, Milli-Q, Millipore, Burlington, MA, USA).

### 2.2. Synthesis of AuNCs

AuNCs were synthesized according to a previously reported method with minor modification [31]. Briefly, an aqueous solution of HAuCl_4_·3H_2_O (20 mM, 0.5 mL) was mixed with 4.35 mL ultrapure water. Then, GSH solution (100 mM, 0.15 mL) was slowly dripped into HAuCl_4_·3H_2_O solutions at 25 °C. After 5 min of magnetic stirring at 500 rpm, the temperature was increased to 70 °C and stirring continued for 24 h. Finally, the AuNCs solution was concentrated to 1.5 times and stored at 4 °C for the following experiments. All glass bottles were washed and soaked in aqua regia before use.

### 2.3. Synthesis of AuNCs@ZIF-8 Nanocomposites

One-pot synthesis of the AuNCs@ZIF-8 nanocomposites was prepared by mixing concentrated AuNCs solution with the precursors of ZIF-8. That is, the Zn(NO_3_)_2_·6H_2_O (0.08 M, 0.5 mL) and 2-MIM (4 M, 3.5 mL) was first dissolved in the concentrated solution of AuNCs solution to obtain solutions A and B, respectively. Then, solution A was dripped into solution B under magnetic stirring (500 rpm) at 25 °C. After 30 min reaction, nanocomposites were sat for 2 h and washed three times under centrifugation (10,000 rpm, 10 min).

### 2.4. Characterization of AuNCs and AuNCs@ZIF-8

Transmission electron microscopy (TEM) images were acquired through a JEOL JEM-2100 transmission electron microscope with an acceleration voltage of 200 kV. UV-vis spectra for AuNCs, ZIF-8 and AuNCs@ZIF-8 in the range of 800–250 nm with scan interval 1 nm were obtained on a T9 UV-vis spectrophotometer provided by Beijing Purkinje General Instrument Co. Beijing China. The crystal structure of ZIF-8 and AuNCs@ZIF-8 were obtained on a D2 PHASER X-ray diffractometer (XRD; Karlsruhe, Germany) using Cu-*K*α radiation as the X-ray source at 30 kV and 10 mA. Fourier transform infrared spectra (FT-IR) in the range of 400–4000 cm^−1^ was measured on an IS10 IR spectrophotometer (Nicolet, Waltham, MA, USA) using KBr pellet method. The fluorescence spectra were observed on a Fluoro Max4 (Horiba JY, Irvine, CA, USA).

To evaluate the fluorescence enhancement effects of AuNCs@ZIF-8, the fluorescence quantum yields of freshly-synthesized AuNCs solution and AuNCs@ZIF-8 synthesized with 1.5 times concentrated AuNCs were measured. Differences in fluorescence intensity of AuNCs and AuNCs@ZIF-8 were additionally measured after dilution to equal AuNCs concentrations. To evaluate the temporal and Ph stability of the prepared AuNCs and AuNCs@ZIF-8, AuNCs and AuNCs@ZIF-8, stored at 4 °C, were taken out every 1 month to measure fluorescence intensity. Changes in fluorescence intensity with Ph of the solutions were measured by respectively taking and diluting 20 μL of AuNCs and AuNCs@ZIF-8 to 1 mL with different Ph of water (pH = 4.5, 5, 6, 7, 8). To investigate the “on-off-enhanced on” fluorescence switch, the fluorescence intensity values of 5 μL of AuNCs@ZIF-8 before and after quenching by Hg^2+^, were recorded, and the fluorescence intensity values were recorded with the addition 1 mL of Cys solution

### 2.5. Optimization of Reaction Conditions

To determine the optimal amount of Hg^2+^, 10 μL of AuNCs@ZIF-8 was added to different volumes of Hg^2+^ solution (3 × 10^−4^ M, 0 μL, 5 μL, 10 μL, 12 μL, 15 μL, 16 μL, 18 μL, 20 μL and 30 μL) and the fluorescence intensity was measured after diluting to 1 mL system with water. To determine the reaction time of AuNCs@ZIF-8 with Hg^2+^, 10 μL of AuNCs@ZIF-8 was added to the optimal amount of Hg^2+^, and the change of fluorescence intensity with time was measured after diluting to 1 mL system with water.

To evaluate the performance of as-prepared nanoprobes, the volume ratio of AuNCs@ZIF-8 to Hg^2+^ ions (3 × 10^−4^ M) was first fixed. Then, the volume of AuNCs@ZIF-8 was changed and the solution systems were replenished to the same volume with ultrapure water. After adding Cys solution (5 μM, 1 mL) and reacting for 15 min, the fluorescence spectrum of the solution was measured under 400 nm excitation. The degree of fluorescence recovery of AuNCs@ZIF-8/Hg^2+^ after the addition of biothiol concentration was expressed by Equation (1):(1)$$\frac{\left(FL-FL_{0}\right)}{FL_{0}}$$

Here, *FL*_0_ and *FL* are the fluorescence intensities of the AuNCs@ZIF-8/Hg^2+^ in the absence and presence of biothiols, respectively.

The optimal reaction time of Hg^2+^/AuNCs@ZIF-8 with Cys was evaluated by adding Cys (1 mL, 5 µM) to Hg^2+^/AuNCs@ZIF-8 and recording the change of fluorescence intensity with time.

### 2.6. Fluorescent Detection of Cys and GSH Molecules

Under shaking, 1 mL of Cys solution at different concentrations (0.0, 1.0, 1.5, 2.0, 2.5, 4.0, 5.0, 8.0, 10.0, 20.0 and 30.0 μM) was separately added into a 5 μL of AuNCs@ZIF-8 solution containing Hg^2+^ ions (3 × 10^−4^ M, 8 μL). Finally, the fluorescence spectra were measured by a fluorescence spectrometer. Parameter settings: excitation wavelength of 500–750 nm, scanning speed at 1 nm/min. Same parameters were used to detect GSH. Moreover, the detection conditions for Cys molecules were optimized accordingly, mainly including dosage of AuNCs@ZIF-8 probe, the dosage ratio of Hg^2+^ ions to AuNCs@ZIF-8 and the incubation time of Cys and Hg^2+^ ions. In addition, the control tests were separately conducted, accordingly, for different analytes (10 folds the Cys solution) of other amino acids (Asp, Arg, Lys, Gly, Val, Ile, Leu, His, Phe, Thr, Pro, Ala, Glu, Trp, Ser, Met, Gln, Asn and Tyr) and GSH. Detection of Cys molecules with AuNCs/Hg^2+^ followed the same detection conditions as AuNCs@ZIF-8/Hg^2+^.

### 2.7. Real Sample Analysis

For the real sample test, bovine serum was first treated with acetonitrile at a volume ratio of 1:5, shaking for 30 s and leaving it to rest for 15 min. After centrifugation to remove protein precipitation at 10,000 rpm for 20 min, acetonitrile was removed by nitrogen blowing and diluted to 200 folds with ultrapure water as solvent. Different concentrations of Cys molecules were pre-prepared with the treated serum for spiked recovery experiments.

## 3. Result and Discussion

### 3.1. Characterization of GSH-AuNCs and AuNCs@ZIF-8

The structure of GSH-AuNCs and AuNCs@ZIF-8 particles were characterized using TEM (Figure 1). As shown in Figure 1A, the original product of GSH-AuNCs particles was a spherical shape with good dimensional uniformity. The mean particle size was 2.21 ± 0.49 nm, calculated from 150 randomly selected particles (Figure 1A insert). Emission fluorescence of GSH-AuNCs showed an optimum at 615 nm under the excitation of 420 nm (Appendix A). Moreover, when the AuNCs solution was irradiated by a UV light at 365 nm, the color of solution changed from a pale yellow to a luminous orange-yellow (Appendix A insert). However, compared with ZIF-8 MOFs, which have a regular uniform polyhedral shape with a uniform size distribution of ca. 120 nm (Figure 1B insert), the one-pot synthesized AuNCs@ZIF-8 particles only showed slight increases in shape and size. Moreover, the zoom-in images, as plotted in Figure 1C, clearly showed that the AuNCs particles were encapsulated inside of the ZIF-8 framework rather than adsorbed on the surface. Meanwhile, UV-vis spectra of AuNCs, ZIF-8 and AuNCs@ZIF-8 particles also indicated that the absorption of AuNCs in AuNCs@ZIF-8 particles was not affected, i.e., the ZIF-8 MOFs had no absorption peak in the range of 250–800 nm, while both AuNCs and AuNCs@ZIF-8 had clear absorption peaks at 400 nm (Appendix A).

The fidelity of the ZIF-8 crystalline structure after imbedding of AuNCs were confirmed with XRD (Figure 2A). The XRD patterns of ZIF-8 and AuNCs@ZIF-8 exhibited characteristic and sharp peaks of pure-phase ZIF-8. It was observed that theAuNCs@ZIF-8 could exhibit no significant change in the phase structures. As plotted in Fourier transform infrared spectroscopy (FT-IR) (Figure 2B), the peaks at 2500–3600 cm^−1^ and 1640–1660 cm^−1^ corresponded to stretching vibrations of -OH groups and C=O stretching in AuNCs, respectively. The characteristic absorption bands of -OH disappeared from the FT-IR spectrum of the AuNCs@ZIF-8 particle. Such obvious variances implied the coordination of -COOH group in GSH with Zn^2+^ ion existing in the ZIF-8 framework. On the other side, an additional characteristic absorption band of AuNCs@ZIF-8 particle at 1640–1660 cm^−1^ also indicated the presence of AuNCs in ZIF-8.

We investigated the fluorescence stability of AuNCs and AuNCs@ZIF-8 with pH and storage time (Figure 2C,D). We found that AuNCs, before and after coating by ZIF-8, did not show significant change in the pH range of 4.5–9. Thus, they presumably had good Ph stability in serum and other detection systems. In addition, the fluorescence intensity of AuNCs and AuNCs@ZIF-8 was not significantly lost during storage of 4 °C for six months.

In addition to the few changes occurring between ZIF-8 and AuNCs@ZIF-8 particles, interestingly, the fluorescence intensity of AuNCs@ZIF-8 (a) was substantially enhanced (approximately sixfold) in comparison to the AuNCs solution (b) with same molarity (Figure 2E insert). It was demonstrated that the enhanced luminescence of AuNCs was linked to the reduced nonradiative decay caused by restriction of intramolecular movements [32]. Thus, it was highly reasonable to speculate that the limited space in ZIF-8 MOFs blocked the intramolecular rotation and provided an unfirm aggregation of AuNCs, which finally resulted in the phenomenon of aggregation-induced emission (AIE) [33,34,35]. The enhanced features were also proven by fluorescence quantum yields and were consistent with previous reports [36]. That is, after optimization of the synthesis conditions of AuNCs@ZIF-8 particles, the quantum yields of primary AuNCs solution and synthesized AuNCs@ZIF-8 particles were 0.75% and 20.65%, respectively. To effectively sense hazards, synthesized AuNCs were concentrated and used at optimum concentration, i.e., 1.5 times in 2-MIM solution (Appendix A).

As plotted in Figure 2F, although the fluorescence intensity of AuNCs@ZIF-8 was intensively improved, it rapidly decreased when Hg^2+^ ions were added. This was probably due to the formation of a metalophulid bond between Hg^2+^ ions and Au^+^ existing on the surface of AuNCs [31,37]. On the other hand, the quenched emissions gradually recovered when the particles met with Cys molecules. Such lost-and-found fluorescence probably originated from the higher, more stable reactions between Hg^2+^ ions and sulfhydryl, compared with the reactions between AuNCs and Hg^2+^ ions [38].

### 3.2. Optimization of Experimental Conditions

To handily control and operate the as-prepared “on-off-enhanced on” approach, the effects of synthesis conditions, including the synthesis time, the volume of GSH solution and the synthesis temperature of fluorescence performances of AuNCs were investigated and the results are plotted in Appendix A. As demonstrated, the fluorescence intensity of AuNCs gradually increased as the reaction kept going, and the optimal intensity was obtained after 24 h (Appendix A). However, when the temperature exceeded 70 °C, the reaction gradually suspended, even though the maximum GSH was set optimally at 150 μL (Appendix A).

In addition to the synthesis conditions, assay operation factors, such as the amount of fluorescence quenching effect observed with Hg^2+^, AuNCs@ZIF-8 composite and the incubation time of biothiols, were also explored to thoroughly elucidate the sensing principles. As shown in Figure 3A, it was observed that the fluorescence intensity gradually decreased with the increase of Hg^2+^ dosage. Such variance implied that more Hg^2+^ could induce a more stable structure of Hg^2+^/Au^+^, but excessive amounts of Hg^2+^ could finally react with biothiols, resulting in a lack of fluorescence response to the target. To obtain optimal detection sensitivity and a wide detection range, the amount of Hg^2+^ was set at 16.0 μL (approximately 90% quenching of initial fluorescence). Under the optimal volume of Hg^2+^, as-prepared probes also displayed an AuNCs@ZIF-8 responding, as illustrated in Figure 3B. With trace numbers of probes, it was difficult to provide good fluorescence; however, an excess of probes undoubtedly reduced the sensitivity of detection. 

Figure 3C,D plotted the time-dependent fluorescence quenching of AuNCs@ZIF-8 to Hg^2+^ ions and fluorescence restoration of the probe of AuNCs@ZIF-8/Hg^2+^ to Cys molecules, respectively. The fluorescence intensity of AuNCs@ZIF-8 particles was significantly quenched in 1 min after meeting with Hg^2+^ ions, and the fluorescence intensity reached a stable level after 5 min. When Cys molecules were present in the solution, the fluorescence intensity of AuNCs@ZIF-8/Hg^2+^ probes sharply increased, reaching the maximum after 10 min. Such a rapid response of our AuNCs@ZIF-8/Hg^2+^ probes demonstrated great potential for onsite POCT detection of biothiols.

### 3.3. Sensitivity and Selectivity of AuNCs@ZIF-8/Hg^2+^ Probes on Cys and GSH Molecules

To investigate the specific features of as-prepared AuNCs@ZIF-8/Hg^2+^ probes on biothiol sensing, the responses of GSH and human amino acids, including Cys molecules, were tested. As plotted in Figure 4, not all of the amino acids displayed evident fluorescence recovery. As is known, Cys molecules are the only amino acid containing -SH groups among the 20 amino acids necessary for humans, while GSH is a thiol-containing tripeptide composed of Cys molecules, glutamic acid and glycine. Therefore, the AuNCs@ZIF-8/Hg^2+^ probes also displayed restored fluorescence on GSH molecules based on interactions between -SH groups and Hg^2+^ ions. However, probably due to the spatial block of tripeptide, the fluorescence intensity of GSH molecules was lower than that of free Cys molecules. The excellent specificity of AuNCs@ZIF-8/Hg^2+^ probes on detection of Cys and GSH molecules suggested that our developed AuNCs@ZIF-8/Hg^2+^ approach exhibited great potential to sense Cys and GSH molecules.

### 3.4. Quantitative Sensing on Trace Amount of Cys and GSH Molecules

The quantitative evaluation of Cys and GSH molecules with our developed AuNCs@ZIF-8/Hg^2+^ approach was studied by measuring the fluorescence intensity at 400 nm. As plotted in Figure 5, after Hg^2+^ was mixed with AuNCs@ZIF-8, it was added to different concentrations of Cys or GHS solutions and incubated for 10 min for fluorescence testing. Cys and GHS molecules were able to establish a series of good linear relationships with AuNCs@ZIF-8-based probes. The detection of Cys molecules with AuNCs@ZIF-8 probes had a linear relationship in the range of 1–10 μM, and the linear equation was y = 2.083x − 2.907 with R^2^ = 0.991. Similarly, the detection of GSH with AuNCs@ZIF-8 probes had a linear relationship in the range of 1.5–8 μM, and the linear equation was y = 0.950x − 1.23 with R^2^ =0.999. Furthermore, compared with previous reports on Cys and GSH molecules, as summarized in Table 1, our AuNCs@ZIF-8/Hg^2+^ strategy displayed a much better response tos trace amount of Cys and GSH molecules regarding linear ranges and LODs, simultaneously.

### 3.5. Detection Performance in Real Samples

To further investigate the feasibility of using AuNCs@ZIF-8/Hg^2+^ probes to sense biothiols in blood sera, recovery experiments ON biothiols in serum samples were carried out. Low, medium and high concentrations of biothiols were selected to verify the universality of our strategy. For the sample preparation, after the protein from bovine serum was precipitated, the supernatant was diluted to a certain concentration, as standard for adding of Cys or GSH molecules, respectively. The results of quantitative evaluation are depicted in Table 2. As listed in Table 2, the recovery of spiked Cys molecules ranged from 97.36% to 115.31%, while that of spiked GSH molecules ranged from 105.29% to 115.46% with an RSD less than 5%. Such quantitative results implied that our developed AuNCs@ZIF-8/Hg^2+^ strategy provided a promising approach to accurate and rapid evaluation of biothiols in blood serum.

## 4. Conclusions

In summary, a “on-off-enhanced on” strategy for sensitive and fast sensing of biothiols in blood serum was developed based on AuNCs@ZIF-8 fluorescence nanoparticles. By combining the ZIF-8 framework with mediators of Hg^2+^ ions, our approach displayed AIE behavior for AuNCs under the embedding of ZIF-8 framework. However, the fluorescence intensity sharply decreased when meeting with mediators of Hg^2+^ ions. Due to the different affinities between the interactions of Hg^2+^ ions with -SH groups and the AuNCs with -SH groups, the quenched probe of AuNCs@ZIF-8/ Hg^2+^ displayed an obviously enhanced fluorescence after the Hg^2+^ ions were competitively interacted with -SH groups. Differing from conventional AIE strategy, our competitively-enhanced AIE behavior resulted in a much lower LOD and longer fluorescence lifetime on sensing trace amounts of biothiols. In light of these findings, this facile, accurate, and reliable competitive AIE strategy could provide a promising alternative method of quantifying trace amounts of biothiols in sera and biological systems.

## Data Availability

The data presented in this study are available on request from the corresponding author.

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
