# Peer review of "A Competitive “On-Off-Enhanced On” AIE Fluorescence Switch for Detecting Biothiols Based on Hg2+ Ions and Gold Nanoclusters"

_biosensors, 2022, doi:10.3390/bios13010035_

Round 1

Reviewer 1 Report

Comments to the manuscript " A Competitive "On-off-Enhanced on" AIE fluorescence switch for detecting biothiols based on Hg2+ ions and gold nanoclusters " The manuscript showed a systematic and interesting study and showed a lot of interesting results. However, I recommended the authors more discussion or description regarding the nanomaterial characterization and the discussion of the results. 

 Please find some additional comments. 

  1. Paraphrase some sentences in the abstract. Long sentence with no sense. Same as in some paragraphs of the manuscript. For example, in line 65. Check the manuscript.
  2. Line 108 replaces “a little” the same as the word "original". It is very subjective.
  3. Please, use commas when appropriate. 
  4. Specify more information in materials and methods. Sample prep, fluorescence conditions, etc.
  5. Please present Equation 1 properly.
  6. Line 177 change configuration for shape.
  7. XRD characterization identifies the peaks and discusses them properly.
  8. Poorly discussion of FTIR
  9. Please, attach the SI file. I am missing the file in the initial submission.

Author Response

Reviewer #1:

Comments to the manuscript " A Competitive "On-off-Enhanced on" AIE fluorescence switch  for detecting biothiols based on Hg2+ ions and gold nanoclusters " The manuscript showed a systematic and interesting study and showed a lot of interesting results. However, I recommended the authors more discussion or description regarding the nanomaterial characterization and the discussion of the results. Please find some additional comments. 

Response: We appreciate very much for the Referee’s comments.

  1. Paraphrase some sentences in the abstract. Long sentence with no sense. Same as in some paragraphs of the manuscript. For example, in line 65. Check the manuscript.

Response: We have carefully checked and corrected the mistakes throughout the whole manuscript.

  1. Line 108 replaces “a little” the same as the word "original". It is very subjective.
    Response: Done. (Page 3 Line 112)

  1. Please, use commas when appropriate.  

Response:  Done 

  1. Specify more information in materials and methods. Sample prep, fluorescence conditions, etc.

Response:  Detailed information and conditions had been list. (Page 3-4 Line 138 - 156)

  1. Please present Equation 1 properly.

Response:  Done. (Page 4 Line 165)

  1. Line 177 change configuration for shape.

Response:  Done (Page 5 Line 203)

  1. XRD characterization identifies the peaks and discusses them properly.

Response: XRD studies on the AuNCs and AuNCs@ZIF-8 was newly performed to verify their chemical features as shown in Fig. 2A. (Page 5 Line 217-220)

  1. Poorly discussion of FTIR

Response: More description of FTIR had been added. (Page 5 Line 220-227)

  1. Please, attach the SI file. I am missing the file in the initial submission

Response: Due to the system transfer, so the SI file was missed.  The new SI file had been added.

Reviewer 2 Report

Pi et al. reported work "A Competitive "On-off-Enhanced on" AIE fluorescence switch 2 for detecting thiols based on Hg2+ ions and gold nanoclusters" is interesting and can be acceptable with minor revision as specified below. 

1) Some recent references related to fluorescent complexes for recognizing biologically relevant ions were missing in the introduction, such as Journal of Inorganic Biochemistry 2020203, 110867, Sens. Actuator B.2019282, 730-742 and Bulletin Korean Chem. Soc. 201940, 163-168. 

2) Is there any interferences from pyrophosphates? 

3) What are the pH effects on the stability of Au nanoclusters?  

4) Some of the typographical mistakes need to be corrected. 

5) FT-IR spectra need to be present according to IUPAC standards, the current presentation (Fig. 2B) is not a conventional one. 

6) Figure 2C inset is not specified with labels; it needs to be corrected.

7) Gold nanocluster's emission stability profile must be provided against a time scale.  

Author Response

Reviewer #2:

Pi et al. reported work "A Competitive "On-off-Enhanced on" AIE fluorescence switch for detecting thiols based on Hg2+ ions and gold nanoclusters" is interesting and can be acceptable with minor revision as specified below. 

  1. Some recent references related to fluorescent complexes for recognizing biologically relevant ions were missing in the introduction, such as Journal of Inorganic Biochemistry 2020, 203, 110867, Sens. Actuator B., 2019, 282, 730-742 and Bulletin Korean Chem. Soc. 2019, 40, 163-168. 

Response:  Done

  1. Is there any interferences from pyrophosphates? 

Response: Based on lots of tests, the interferences from pyrophosphate did not observe.

3.What are the pH effects on the stability of Au nanoclusters?  

Response: The effects of pH on stabilities of AuNCs and AuNCs@ZIF-8 were shown in Fig. 2C and 2D. (Page 5 Line 228-233, Page 6 Fig. 2)

  1. Some of the typographical mistakes need to be corrected. 

Response:  Done.  Please find marked areas with Red color.

  1. FT-IR spectra need to be present according to IUPAC standards, the current presentation (Fig. 2B) is not a conventional one. 

Response: The corrections had been given as explained in Page 6 Line 249 and Fig. 2B.

  1. Figure 2C inset is not specified with labels; it needs to be corrected.

Response: The label information had been stated. Please find Page 6 Fig. 2E.

  1. Gold nanocluster's emission stability profile must be provided against a time scale.

Response: A stability of AuNCs and AuNCs@ZIF-8 over a six-month period had been shown in Fig. 2D. Please find the Page 5 Line 228-233.

Reviewer 3 Report

The submitted manuscript “A Competitive "On-off-Enhanced on" AIE fluorescence switch 2

for detecting biothiols based on Hg2+ ions and gold nanoclusters” by Li et.al. can not be published in this journal. I would recommend to publish such a observation based result in a letter format or perform a significant improvement.

1. Abstract should be more refined and a few sentences are not evident in the first reading.

2. Introduction looks fine. Would authors mind to discuss the hole sizes in ZIF in case of encapsulation of molecules or nanoparticles or cluster of nanoparticles, they should provide a relationship. A clear motive to use good clusters or aggregation is not visible. How is this AIR-based luminescence is better? What would happen for trace elements recognition using the same or similar technique?

3. Is there any reference for synthesis? Can authors do a surface characterization such as SEM to confirm the cluster shape? compared XRD with database would help to see the crystallized size then its relation with emission. I would suggest the particle size distribution in the leading figure 1. FFT analysis can help to understand the phase of Au and ZIF structures. 

4. Authors should improve Figure 2b  by tagging the peaks, and it is a careless plot; it is FTIR, not absorbance, and the unit on the x-axis is cm-1. Mark line 204. Should explain the interaction mechanism.

5. In figure 2 c and d, please differentiate red in c, and black in d. 

This paper lacks scientific information on AIE-based results but discusses observation-based results. Please explain the mechanism of the “on-off-enhanced on” strategy for sensitive and fast sensing of bio-thiols in blood serum? Figure quality and references must be checked and verified. 

Author Response

Reviewer #3:

The submitted manuscript “A Competitive "On-off-Enhanced on" AIE fluorescence switch for detecting biothiols based on Hg2+ ions and gold nanoclusters” by Li et.al. can not be published in this journal. I would recommend to publish such a observation based result in a letter format or perform a significant improvement.

  1. Abstract should be more refined and a few sentences are not evident in the first reading.

Response: The English had been thoroughly corrected.

  1. Introduction looks fine. Would authors mind to discuss the hole sizes in ZIF in case of encapsulation of molecules or nanoparticles or cluster of nanoparticles, they should provide a relationship. A clear motive to use good clusters or aggregation is not visible. How is this AIR-based luminescence is better? What would happen for trace elements recognition using the same or similar technique?

Response: Actually, the present study is not predesigned based on ZIF. This work is mainly focusing on the sensitive detection of biothiols. As explained at the first sentence in abstract and the last paragraph in the introduction, the present work would like to propose a new approach for sensitive detection on trace biothiols based on AIE phenomenon. 

  1. Is there any reference for synthesis? Can authors do a surface characterization such as SEM to confirm the cluster shape? compared XRD with database would help to see the crystallized size then its relation with emission. I would suggest the particle size distribution in the leading Fig. 1. FFT analysis can help to understand the phase of Au and ZIF structures. 

Response: Detailed methods for developing the particles and related nanocomposites had been given in Section 2.

The particle size had been shown in Fig. 1A insert, and the detailed explanation about structures of AuNCs and ZIF were given in lines of 201-216 Page 5.

  1. Authors should improve Figure 2b by tagging the peaks, and it is a careless plot; it is FTIR, not absorbance, and the unit on the x-axis is cm-1. Mark line 204. Should explain the interaction mechanism.

Response: Done. Please find new figure of Fig. 2B and the explanations of line 220-227.

By the way, the FTIR spectroscopy could be plotted as Transmittance and Absorbance. Compared with Transmittance type of spectroscopy, the absorbance type of spectroscopy could provide information about the concentration of functional groups following Lamber-Beer’s law. Therefore, currently, lots of researchers would like to employ Absorbance instead of Transmittance in FTIR spectroscopy.

  1. In figure 2 c and d, please differentiate red in c, and black in d. 

Response: Done. Please find Fig. 2. (Page 6 Fig. 2E)

This paper lacks scientific information on AIE-based results but discusses observation-based results. Please explain the mechanism of the “on-off-enhanced on” strategy for sensitive and fast sensing of bio-thiols in blood serum? Figure quality and references must be checked and verified. 

Response: Thanks for the reviewer’s comment. However, it is hard for us to fully agree with his/her subjective views. Our present work mainly focuses on the application of AIE phenomenon for sensitive sensing biothiols, the materials construction is not the topic. Therefore, to easily understand our novel strategy, the material characterizations should not be put too much.

Round 2

Reviewer 1 Report

The authors improve the quality of the article. However, I have a few more comments.

Figure 2S, please include 0M AuNCs solution and 0 M 2-MIM.

Figure 3S, please include in the graph volume of glutathione 0 uL.

Table 1, missing control experiments. Cys and GHS 0uM.

Figure 3B, what is the x axe?

Figure 2, FTIR, changes the x axe. For better guidelines and visualization, Indicate the functional groups in the graph. If it is too small, the authors can move the SI file.

Author Response

The authors improve the quality of the article. However, I have a few more comments.

 Response: We appreciate very much for the Referee’s comments.

  1. Figure 2S, please include 0M AuNCs solution and 0 M 2-MIM.

 Response: The data that the dosage of 2-MIM is 0 M has been supplemented. When the dosage of AuNCs is 0 M, the material is ZIF-8 and there is no fluorescence. (SI Fig.S2A)

  1. Figure 3S, please include in the graph volume of glutathione 0 uL.

  Response: When the volume of glutathione is 0 uL, AuNCs cannot be synthesized.

  1. Table 1, missing control experiments. Cys and GHS 0uM.

  Response: The two figures in the second column of Table 2 are the values of 0 μM for the addition of Cys and GSH in the labeling experiment. That is, 1.008 μM and 1.567 μM are the detection values of after adding only diluted serum to Hg2+/AuNCs@ZIF-8. If only water is added instead of diluted serum, there is no fluorescence recovery in the system.

  1. Figure 3B, what is the x axe?

  Response: The X-axis is the amount of AuNCs@ZIF-8 for the detection of b biothiols.

  1. Figure 2, FTIR, changes the x axe. For better guidelines and visualization, Indicate the functional groups in the graph. If it is too small, the authors can move the SI file.

 Response: We have made changes in the article. (Fig.2B)

Reviewer 3 Report

Li et al. have significantly improved the quality of the manuscript and responded to the reviewer's concerns quite well. I would recommend the publication with minor changes in this journal. I would ask the authors to improve the text in the scheme, Figure 1a, 2b, x-axis of Figure 4, and some English mistakes. 

Author Response

Reviewer #3:

Li et al. have significantly improved the quality of the manuscript and responded to the reviewer's concerns quite well. I would recommend the publication with minor changes in this journal. I would ask the authors to improve the text in the scheme, Figure 1a, 2b, x-axis of Figure 4, and some English mistakes. 

Response: Thank you for your suggestion. We have made relevant changes in the article.